

# Behind the wheel: exploring gray matter variations in experienced drivers

Jiangtao Chen[1,*], Xiaoyu Chen[1,*], Li Gong[1], Di Zhang[2] and Qiang Liu[1,3]

[1] Research Center of Brain and Cognitive Neuroscience, Liaoning Normal University, Dalian, Liaoning, China
[2] School of Psychology, Guizhou Normal University, Guiyang, Guizhou, China
[3] Institute of Brain and Psychological Sciences, Sichuan Normal University, Chengdu, Sichuan, China
* These authors contributed equally to this work.

Corresponding author
Qiang Liu, lq780614@163.com

## ABSTRACT

**Background:** Driving is a complex skill involving various cognitive activities. Previous research has explored differences in the brain structures related to the navigational abilities of drivers compared to non-drivers. However, it remains unclear whether changes occur in the structures associated with low-level sensory and higher-order cognitive abilities in drivers.
**Methods:** Gray matter volume, assessed *via* voxel-based morphometry analysis of T1-weighted images, is considered a reliable indicator of structural changes in the brain. This study employs voxel-based morphological analysis to investigate structural differences between drivers ($n = 22$) and non-drivers ($n = 20$).
**Results:** The results indicate that, in comparison to non-drivers, drivers exhibit significantly reduced gray matter volume in the middle occipital gyrus, middle temporal gyrus, supramarginal gyrus, and cerebellum, suggesting a relationship with driving-related experience. Furthermore, the volume of the middle occipital gyrus, and middle temporal gyrus, is found to be marginally negative related to the years of driving experience, suggesting a potential impact of driving experience on gray matter volume. However, no significant correlations were observed between driving experiences and frontal gray matter volume.
**Conclusion:** These findings suggest that driving skills and experience have a pronounced impact on the cortical areas responsible for low-level sensory and motor processing. Meanwhile, the influence on cortical areas associated with higher-order cognitive function appears to be minimal.

## INTRODUCTION

Driving, as a daily skill, requires the highly coordinated execution of complex cognitive activities. Behavioral studies indicate that experienced drivers, compared to non-drivers or novice drivers, demonstrate advantages in perceiving the road environment, particularly in challenging road conditions. They exhibit the ability to timely perceive potential hazards and take appropriate measures promptly. This proficiency extends beyond fundamental

cognitive functions such as attention and motor responses, encompassing higher cognitive functions like decision-making (*Crundall, 2016*; *Horswill, 2016*; *Smith et al., 2009*; *Ventsislavova et al., 2016*). The paramount importance of driving skills in traffic safety has propelled extensive neuroimaging research on the subject (*Just, Keller & Cynkar, 2008*; *Spiers & Maguire, 2007*; *Uchiyama et al., 2003*; *Walter et al., 2001*).

Building upon the recognition of the cognitive complexities involved in driving, *Maguire et al. (2000)*, *Maguire, Woollett & Spiers's (2006)* and *Woollett & Maguire (2011)* conducted a series of experiments utilizing voxel-based morphological analysis (VBM) to investigate the neuroanatomical correlates of licensed drivers in London, both cross-sectionally and longitudinally. Their research revealed significant morphological changes in the hippocampus of London taxi drivers compared to non-drivers. Given the close association between the hippocampus and spatial navigation-related cognitive activities, they attributed this finding to corresponding changes in the brain cortex plasticity of drivers resulting from their long-term experience with the intricate road layouts in the local environment.

However, driving, as a skill that requires the simultaneous involvement of various cognitive activities, is not exclusively associated with spatial navigation cognitive abilities related to road layouts. Clearly, experienced drivers excel in handling various unexpected road situations (*Horswill, 2016*; *Ventsislavova et al., 2016*). Therefore, compared to familiarity with road layouts, the ability to perceive, interpret, and respond timely to hazardous events in complex road conditions is more crucial. Neuroimaging studies support this notion, indicating that visual processing, auditory processing, sensorimotor coordination, decision-making, and other low-level or higher-level cognitive activities should also be improved with enriched driving experience (*Spiers & Maguire, 2007*; *Wang et al., 2015*). A meta-analysis study further reveals activation in the pre-motor cortex, occipital visual regions, occipitoparietal regions, and temporal gyrus regions during driving (*Lappi, 2015*). In summary, driving is a complex task involving multiple cognitive functions; thus, considering cortical plasticity, long-term driving not only leads to changes in cortical regions related to spatial-related functions but also induces extensive structural changes in the brain due to the involvement of other cognitive activities.

However, previous investigations into changes in the brain morphology due to driving experience have not thoroughly explored whether, beyond spatial navigation abilities, other cognitive activities also undergo morphological changes with enriched driving experience. To explore the brain morphological changes induced by driving experience, particularly the impact on brain regions associated with sensory processing and higher cognitive activities, this study will employ MRI technology to assess the gray matter volume of both drivers and non-drivers. Our specific aim is to identify the brain regions undergoing morphological changes in drivers due to driving experience by comparing the gray matter volumes between drivers and non-drivers. We will focus on investigating changes in the cortex responsible for sensorimotor processing and the frontal cortex responsible for higher-order cognitive control resulting from long-term driving.

Previous neuroimaging studies have indicated that the act of driving engages multiple brain regions, including the pre-motor cortex, occipital visual regions, occipitoparietal

regions, and temporal gyrus regions in virtual simulated driving (*Lappi, 2015*; *Uchiyama et al., 2003*; *Walter et al., 2001*). Although these identified brain regions are closely related to sensorimotor functions, they may not necessarily be linked to higher-order cognitive functions such as decision-making and executive control, which typically involve frontal cortex. This suggests that engaging in long-term driving may not necessarily lead to the enhancement of higher cognitive functions. It is important to note, however, that the relationship between functional activation and changes in brain morphology is not always straightforward (*Baria et al., 2013*; *Park & Friston, 2013*). Therefore, we will further investigate changes in brain structure to determine whether driving skills primarily improve low-level sensorimotor processing abilities, higher-order cognitive control capabilities, or a combination of both.

In addition, numerous studies suggest that individuals mastering specialized skills often undergo profound changes in specific cortical and subcortical brain structures due to neuroplasticity. These changes are diverse, encompassing not only static alterations but also sustained modifications in neural structures associated with the prolonged learning and practice of a particular skill (*Münte, Altenmüller & Jäncke, 2002*; *Schlaug, 2001*; *Wang et al., 2020*). In other words, for a specialized skill, the processing efficiency of certain brain regions can continue to improve even after proficiency in the skill is achieved. These enduring structural changes in the brain that persist after acquiring a skill are evidently experience-related. To further address the question of which brain regions associated with driving skills undergo sustained changes with increasing experience and which do not, we will conduct correlational analyses between the gray matter volume of these driving-specific brain regions and the driving experience, measured by the driver's years of driving experience.

## MATERIALS AND METHODS

### Subjects

A total of 42 participants took part in this paid experiment conducted in Beibei District, Chongqing, China (see Table 1 for demographic details). The participant pool consisted of 22 licensed drivers (21 males, mean age: 40.4 ± 6.07 years) and 20 non-drivers (18 males, mean age: 42 ± 1.89 years). Age and education backgrounds were carefully matched between the two groups. All participants were right-handed and recruited based on specific criteria to ensure uniformity. Licensed drivers had an average driving experience of 11.6 years (ranging from 6 to 23 years), while non-drivers had no prior knowledge of driving. To maintain homogeneity, participants across both groups had no history of major head trauma, alcohol or drug addiction, or any neurological disorders. Additionally, all subjects provided written informed consent and were kept unaware of the experiment's purpose to prevent potential biases. The study was approved by the Institutional Review Board of the Brain Imaging Center at the Southwest University (SWU) (the approval number: H09025). The methods employed in this study followed the approved guidelines to ensure ethical conduct throughout the research.

**Table 1 The characteristics of the participants recruited in this study.**

| Variable | Driver | Non-driver | $t$-value | $p$ value |
|---|---|---|---|---|
| Sample size | 22 | 20 | | |
| Age (years) | 40.4 ± 6.07 | 42 ± 1.89 | −1.123 | 0.268 |
| Gender (male/female) | 21/1 | 18/2 | $\chi^2 = 0.47$ | 0.493 |
| Education (years) | 9.54 ± 1.76 | 9.0 ± 1.34 | $\chi^2 = 3$ | 0.223 |
| Total driving (years) | 11.63 ± 4.76 | | | |

## Data acquisition

During the MRI scanning sessions, participants were given explicit instructions to maintain a motionless state, close their eyes, and remain awake throughout the procedure. The imaging data were acquired using the SIEMENS TRIO 3T MRI scanner at Southwest University. High-resolution T1-weighted anatomical images were obtained with the following parameters: repetition time (TR) of 2,530 ms, echo time (TE) of 3.39 ms, flip angle of 70°, field of view (FOV) measuring $256 \times 256$ mm$^2$, and a slice thickness of 1 mm. These imaging parameters were carefully selected to ensure optimal data quality and resolution.

## Data analysis

### MRI quality control

Before initiating preprocessing steps, a rigorous initial visual quality assessment was conducted on all images. This assessment targeted potential issues related to motion, gross anatomical artifacts, and ensured comprehensive whole-brain coverage. Following preprocessing, additional insights into data quality, including resolution, noise, and bias, were obtained through CAT12 (http://www.neuro.uni-jena.de/cat, cat12.8_r1871) (*Gaser et al., 2022*) and SPM12 (https://www.fil.ion.ucl.ac.uk/spm/, v7771). A weighted average quality score of B or higher, indicative of very good image quality, was confirmed for all datasets. To further ensure data reliability, a post-preprocessing quality check was performed using the CAT12 toolbox. This involved assessing sample homogeneity by examining standard deviations before proceeding to the subsequent statistical analyses. These stringent quality control measures contribute to the robustness of our data, affirming the integrity and consistency of the neuroimaging results.

### Processing

High-resolution structural images for each participant were processed using CAT12 (cat12.8_r1871) implemented in SPM12 to estimate gray matter volume with default parameters. CAT12 autonomously conducted intra-subject realignment, bias correction, segmentation, and normalization. Segmentation involved classifying brain volumes into three voxel categories: gray matter (GM), white matter (WM), and cerebrospinal fluid (CSF), using adaptive maximum *a posteriori* segmentation and partial volume segmentation. Extracted GM maps underwent smoothing with an 8 mm full-width at half-maximum (FWHM) kernel. Group-wise differences in GM volumes were examined

**Table 2 Comparison between drivers and non-drivers in the volume of the gray matter, white matter, cerebrospinal fluid, and whole brain.**

|  | Drivers (ml) | Non-drivers (ml) | t | p |
|---|---|---|---|---|
| Gray matter | 661.52 ± 54.88 | 674.9 ± 43.55 | −0.869 | 0.390 |
| White matter | 536.89 ± 48.17 | 537.56 ± 49.89 | −0.045 | 0.965 |
| Cerebrospinal fluid | 266.57 ± 37.85 | 251.81 ± 35.16 | 1.306 | 0.199 |
| Whole brain | 1,465 ± 103.59 | 1,464.28 ± 96.96 | 0.023 | 0.982 |

through a whole-brain analysis using a two-sample t-test within the general linear model. Covariates, including age, gender, and total intracranial volume (TIV), were controlled. Statistical significance was assessed using the threshold-free cluster enhancement (TFCE) method (*Salimi-Khorshidi, Smith & Nichols, 2011*; *Smith & Nichols, 2009*) for family-wise error (FWE) correction ($p < 0.05$).

To establish region-specific associations, significant clusters from the whole-brain analysis were used to create region of interest (ROI) masks. Mean values within these masks were extracted for each individual using the get_totals function (http://www0.cs.ucl.ac.uk/staff/g.ridgway/vbm/get_totals.m) for SPM. Associations between voxel-based morphometry (VBM) results and driving years in these ROIs were assessed using partial correlation, with TIV, age, and education age as covariates in SPSS 22.0.

## RESULTS

### Global volumes

No significant differences were found between drivers and nondrivers in the volume of the global MG, WM, and CSF. For details see Table 2.

### Whole-brain analysis

Significant differences in gray matter (GM) volumes between drivers and non-drivers were observed (see Fig. 1 and Table 3). The results indicated that drivers exhibited significantly smaller GM volumes than non-drivers.

### ROI analysis

The ROI analysis revealed significant volumetric differences in specific brain regions (see Fig. 2). Specifically, Drivers displayed a smaller gray matter volume than nondrivers in regions including the left middle temporal gyrus ($t (40) = -3.607$, $p = 0.001$ Cohen's $d = -1.122$), right middle occipital gyrus ($t (40) = -3.588$, $p = 0.001$, Cohen's $d = -1.116$), right cerebellum_4_5 ($t (40) = -2.234$, $p = 0.031$, Cohen's $d = -0.692$) and right supramarginal gyrus ($t (40) = -2.994$, $p = 0.005$, Cohen's $d = -0.925$).

### The relationship between the volume of ROIs and the years of driving experience

A marginally significant negative correlation was found between the years of driving experience and the volumes of the left middle temporal gyrus ($r = -0.418$, $p = 0.075$,
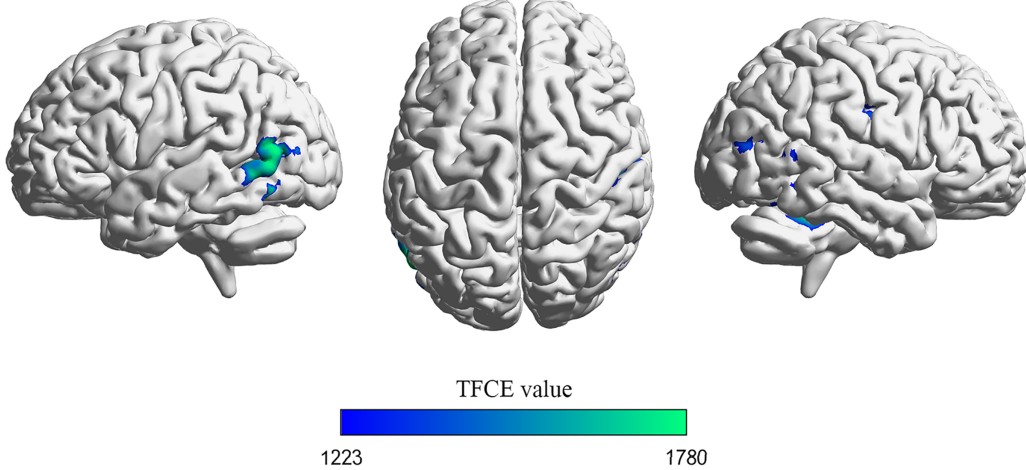

**Figure 1 Voxel-based morphometry (VBM) analysis results.** Brain regions exhibiting a significant decrease in gray matter volume in drivers compared to non-drivers are depicted. The maps were thresholded using threshold-free cluster enhancement (TFCE) at *p* < 0.05, corrected for family-wise error (FWE) across the whole brain. Visualization of brain space graphs was performed using BrainNet Viewer (http://www.nitrc.org/projects/bnv/) (*Xia, Wang & He, 2013*). TFCE statistics are represented by the color bar.                                   

**Table 3 Scatter plots illustrating correlations between ROI volumes and driving years.**

| Region | BA | Cluster size (mm³) | MNI coordinate (peak) | | | *p* (peak) | TFCE value |
|---|---|---|---|---|---|---|---|
| | | | **x** | **y** | **z** | | |
| MTG_L | 37 | 1,950 | −57 | −71 | 11 | 0.01 | 1,779.77 |
| Cerebellum_4_5_R | / | 1,641 | 26 | −44 | −20 | 0.012 | 1,728.13 |
| SMG_R | 3 | 933 | 56 | −24 | 41 | 0.019 | 1,554.51 |
| MOG_R | 19 | 491 | 47 | −81 | 18 | 0.029 | 1,410.60 |

**Note:**
*p* < 0.05 FWE corrected with threshold-free cluster enhancement (TFCE). BA (Brodmann area). MTG_L (left middle temporal gyrus), Cerebellum_4_5_R (right cerebellum 4_5), SMG_R (right supramarginal gyrus), MOG_R (right middle occipital gyrus).

Fig. 3A), the right middle occipital gyrus (r = −0.414, *p* = 0.078, Fig. 3B). However, the correlation between the volume of right cerebellum_4_5 (r = −0.395, *p* = 0.095, Fig. 3C), as well as the right supramarginal gyrus (r = −0.007, *p* = 0.978, Fig. 3D) and the years of driving experience, did not reach significance.

## DISCUSSION

In this study, we explored the brain morphological changes induced by driving experience, particularly in the brain regions associated with sensory and higher-level cognitive processing. Building upon these analyses, we investigated whether these relevant brain regions undergo sustained changes associated with enriched experience. Our results indicate that, compared to non-drivers, drivers exhibit smaller gray matter volumes in the middle occipital gyrus, middle temporal gyrus, supramarginal gyrus, and cerebellum_4_5. Notably, no significant gray matter volume changes related to driving were observed in the frontal cortex. This suggests that driving skills have a pronounced impact on the cortical

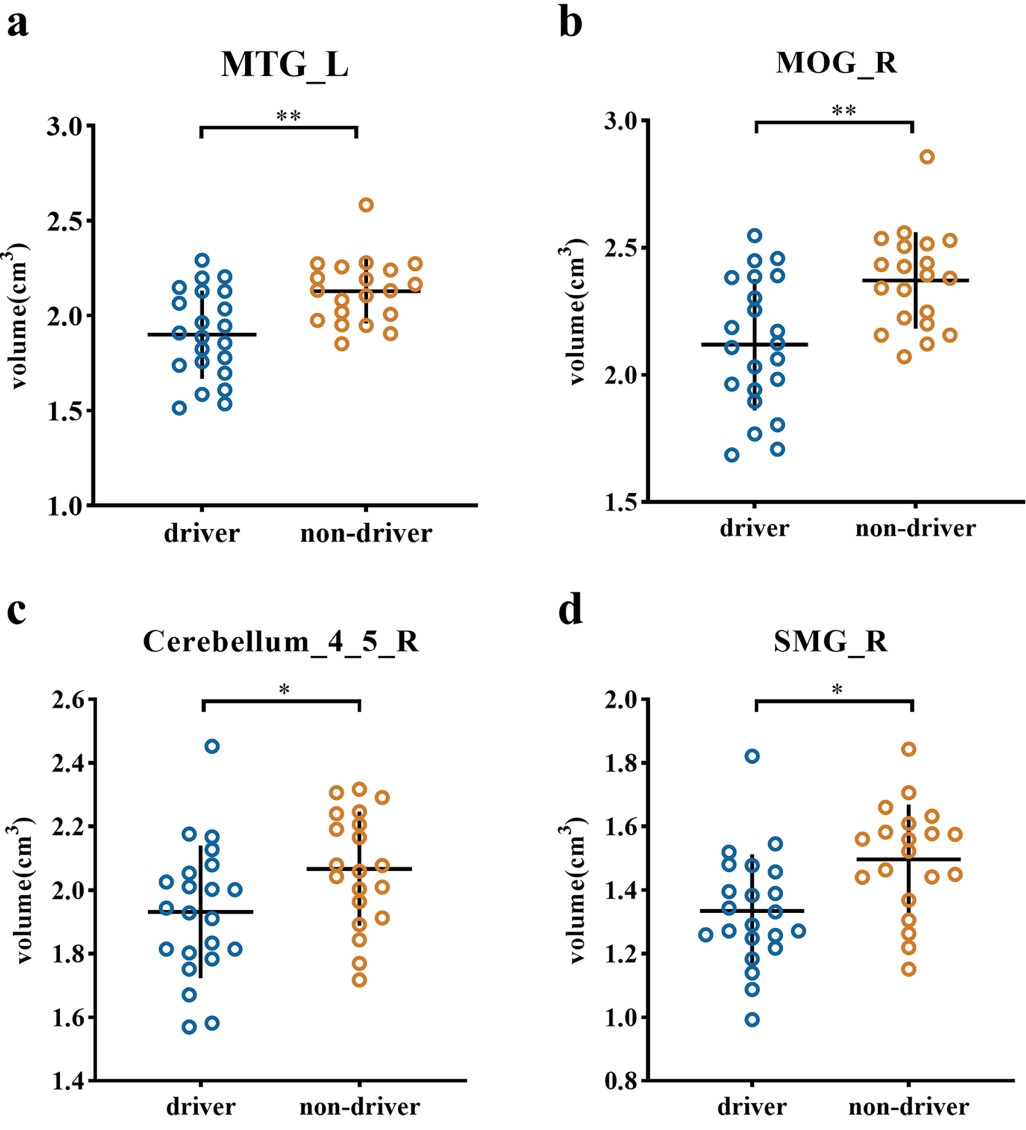

**Figure 2 Scatter plot of individual volume in the ROIs between driver and non-driver.** Individual volumes in specific regions of interest (ROIs) are presented: (A) MTG_L, (B) MOG_R, (C) cerebrellum_4_5_R and (D) SMG_R. The horizontal lines represent the mean of each group while the error bars represent the standard deviation (SD). The asterisks (*) at the top of graph denote *p < 0.05, **p < 0.01; Note: MTG_L (left middle temporal gyrus), MOG_R (right middle occipital gyrus), cerebellum_4_5_R (right cerebellum_4_5), SMG_R (right supramarginal gyrus).

areas responsible for low-level sensory and motor processing, while the influence on cortical areas associated with high-level cognitive functions is minimal. Additionally, we found a marginally negative correlation between the gray matter volumes of the right middle occipital gyrus, left middle temporal gyrus with years of driving experience. However, no significant gray matter volume changes related to driving experience were observed in the supramarginal gyrus and cerebellum_4_5. This implies that the gray matter volumes of the right middle occipital gyrus and left middle temporal gyrus are

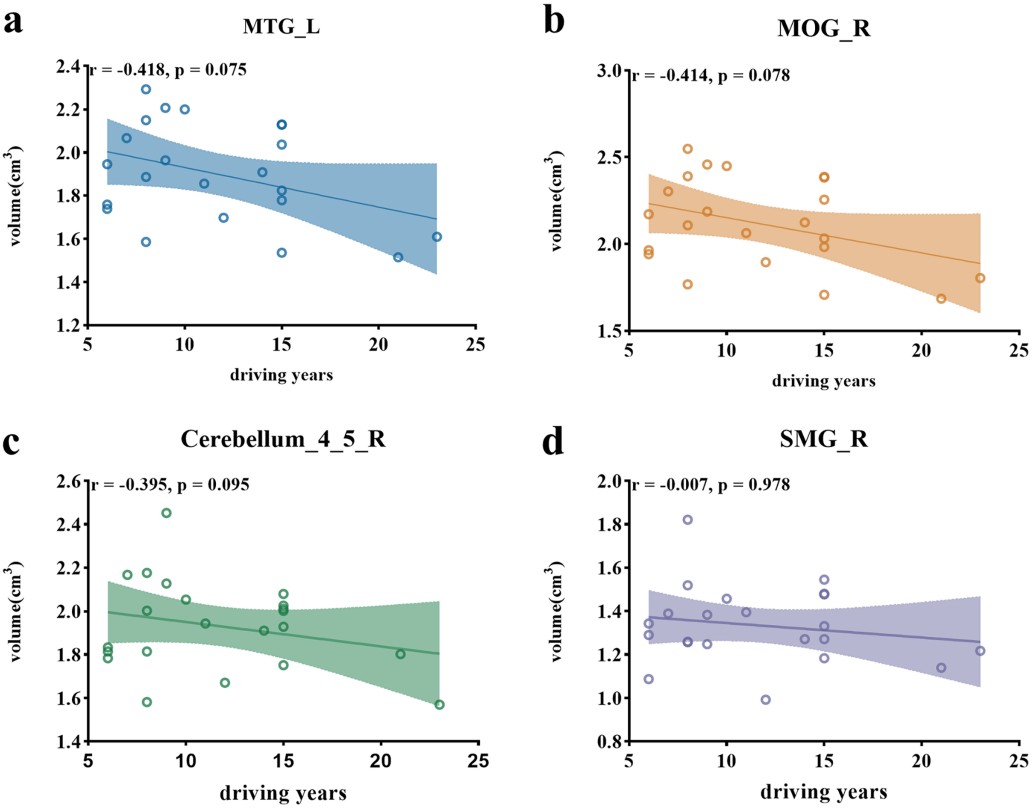

**Figure 3 Scatter plots illustrating correlations between ROI volumes and driving years.** (A) Scatter plot depicting the negative correlation between the volume of the left middle temporal gyrus and driving years. (B) Scatter plot illustrating the negative correlation between the volume of the right middle occipital gyrus and driving years. (C) Scatter plot showing the correlation between the volume of the right cerebellum_4_5 and driving years. (D) Scatter plot presenting the correlation between the volume of the right supramarginal gyrus and driving years. Note: MTG (left middle temporal gyrus), MOG_R (right middle occipital gyrus), cerebellum_4_5_R (right cerebellum_4_5), SMG_R (right supramarginal gyrus).

associated with driving experience, confirming that the key difference brought about by driving experience lies in the efficiency of low-level sensory.

Previous research indicates that extensive learning and practice of specialized skills lead to notable changes in brain structures associated with those skills (*Draganski et al., 2004*; *Maguire et al., 2000*; *Münte, Altenmüller & Jäncke, 2002*; *Zou et al., 2012*). During the learning process, the brain undergoes profound anatomical changes, such as alterations in cell size, growth, or atrophy of neurons or glial cells, due to increased efficiency in neural pathways associated with specific cognitive activities (*Anderson, 1980*; *Duerden & Laverdure-Dupont, 2008*). For instance, musicians exhibit significant gray matter volume changes in sensorimotor-related regions (*Münte, Altenmüller & Jäncke, 2002*; *Schlaug, 2001*), while chess players show adaptability in the thalamus (*Duan et al., 2014*; *Wang et al., 2020*). However, the relationship between skill acquisition and cortical volume remains contentious. Although the prevailing view suggests a positive correlation between the duration of skill practice and cortical gray matter volume, *i.e.*, "more skill, more gray matter volume" (*Hyde et al., 2009*; *Münte, Altenmüller & Jäncke, 2002*; *Sluming et al.,*
*2002*), some studies present contradictory conclusions. Numerous studies found that training and expertise might lead to local decreases in cortical volume, *i.e.*, "more skill, less gray matter volume" (*Granert et al., 2011*; *Hänggi et al., 2010*, *2014*). The former implies that the brain enhances efficiency in cognitive activities related to a particular skill by increasing the number of neurons and synapses associated with skill-related cognitive processing (*Draganski et al., 2004*; *Hyde et al., 2009*; *Sluming et al., 2002*). In contrast, the latter suggests that reduced gray matter volume was results from an increasing proportion of myelinated axons in white matter during structural changes of the human brain, as demonstrated in the myelination model (*Whitaker et al., 2016*). For example, previous study has found that the occipital and temporal lobes white matter increased myelination and lead to gray matter reduction (*Megías et al., 2018*).

Consistent with the myelination model, our study did not find any brain regions that increased gray matter volume with mastery of driving skills or increased driving experience. Instead, we observed significant decreases in gray matter volume in brain cortical regions related to driving skills. This suggests that, at least for driving skills, the brain improves efficiency in related cognitive activities by decreasing proportion of gray matter in the occipitotemporal cortex. Thus, our results support the notion of "more skill, less gray matter volume," from the myelination model.

The supramarginal gyrus is a crucial component of the somatosensory association cortex, playing a vital role in multisensory integration, action reprogramming, tool-use actions, and predictive motion planning and responses. It is commonly considered responsible for processing sensorimotor information (*Hartwigsen et al., 2012*; *Lesourd et al., 2017*; *McDowell et al., 2018*; *Potok et al., 2019*) and undergoes plastic changes with movement-related experiences (*Elbert & Rockstroh, 2004*; *Jäncke, Shah & Peters, 2000*; *Jäncke, 2009*). *James et al. (2014)* reported a reduction in gray matter volume in the sensorimotor region among expert musicians, suggesting neuroplastic splicing in response to increased demands of multisensory integration in motor skills. Similarly, our study observed a comparable phenomenon, where proficiency in driving skills resulted in decreased gray matter volume in the supramarginal gyrus. Importantly, we further discovered that the extent of this movement-related gray matter volume reduction is limited and does not continually decrease with prolonged driving experience. This result indicates that, as a daily skill requiring complex cognitive processing, there is a ceiling to the improvement in cognitive processing efficiency related to bodily movements with increasing driving experience (*Dayan & Cohen, 2011*). The key factor contributing to variations in driving abilities among drivers is not whether they can smoothly execute movements like pressing the accelerator or brake pedals but rather the preceding cognitive processing involved in these movement-related skills.

Similar to the supramarginal gyrus, the cerebellum_4_5 is also a region associated with movement-related functions. Previous research indicates that the cerebellum integrates multisensory information from the somatosensory cerebellar hemispheres to compute a "state estimate," which is crucial for accurate action planning and optimization (*Bhanpuri, Okamura & Bastian, 2013*; *Imamizu et al., 2000*). Recent studies have identified a reduction in specific lobule gray matter volume associated with drum training and musical

instrument usage (*Baer et al., 2015*; *Bruchhage et al., 2020*). Clearly, these musical skills require coordination of movement speed and integration of temporal-motor information. Prior research suggests the involvement of the cerebellum in fine control processes during driving-related motor execution based on predetermined motivations (*Spiers & Maguire, 2007*), emphasizing the cerebellum's necessity in coordinating movement speed and temporal-motor integration for driving behavior (*Hung et al., 2014*). Thus, the observed reduction in gray matter volume in the cerebellum_4_5 in our study likely signifies substantial changes in driving-related movement preparation and coordination control abilities resulting from mastering driving skills. Unfortunately, despite the association between cerebellum_4_5 and driving skills, its gray matter volume does not change with increasing driving experience. Therefore, the differential impact of driving experience does not seem to stem from alterations in movement preparation and coordination control.

In summary, evidence from gray matter volume suggests that improvements in movement-related skills related to driving reach a relatively rapid ceiling. This raises a new question regarding the enhancements in cognitive functions implied by extensive driving experience. The wealth of experiential knowledge gained through driving signifies a heightened proficiency in promptly responding to diverse situations on the road. This is particularly salient within the intricate milieu of the Chongqing highway system, replete with a complex road network and a multitude of vehicles. Evidently, the richness of driving experience not only culminates in the acquisition of a substantial repertoire of driving-related motor skills but also necessitates more streamlined sensory processing. This, in turn, engenders heightened efficacy in situation analysis and facilitates decisive decision-making.

In addition to the SMG and the cerebellum_4_5, we observed marginal reductions in gray matter volume in the middle occipital gyrus and middle temporal gyrus, both associated with driving experience. Importantly, the extent of gray matter volume reduction in these two brain regions does not seem to reach a specific limit over an extended period but rather diminishes with increasing driving experience. Previous research has confirmed the involvement of the middle occipital gyrus and middle temporal gyrus in various cognitive processes, notably including visual perception, auditory perception, and audio-visual cross-modal sensory integration (*Duan et al., 2014*; *Fan et al., 2017*; *Weng et al., 2017*). Prior studies have established that learning a particular task leads to a reduction in task-related activation in these brain regions (*Marois, Leung & Gore, 2000*; *Summerfield & Egner, 2009*). Therefore, the reduction in gray matter volume in these two brain regions may be associated with more efficient sensory perception.

Resting state fMRI studies related to driving have consistently reported a significant reduction in brain activity within the visual network of drivers when compared to non-drivers (*Wang et al., 2015*). Building upon this evidence, we propose that the observed decrease in gray matter volume in the middle occipital gyrus signifies a streamlining of neural pathways dedicated to processing visual information, ultimately enhancing the efficiency of visual signal transmission (*Marois, Leung & Gore, 2000*; *Summerfield & Egner, 2009*). This structural adaptation in the middle occipital gyrus aligns logically with the demands of driving. During this activity, the swift and accurate analysis of information

within the field of view, coupled with the rapid transmission of this information to higher brain regions, forms the basis for effectively navigating unexpected situations. Notably, one study suggests that, relative to non-drivers, drivers exhibit decreased spontaneous brain activity in the visual and sensory networks. Another fMRI investigation indicates weakened functional connectivity between the left posterior medial entorhinal cortex (pmEC) and the right angular gyrus, bilateral precuneus, and some temporal regions in drivers (*Peng et al., 2018*). When considered alongside our findings, this evidence suggests that this weakened functional connectivity may be indicative of active neural pruning in relevant brain regions. A study by *Vaquero et al. (2016)* found that piano-playing experts, compared to novices, have smaller gray matter volume in the auditory cortex, further reinforcing our assertion.

It is noteworthy that we did not observe changes in gray matter volume in the frontal cortex, specifically associated with higher-order cognitive function such as decision-making or executive control, due to driving skills or driving experience. This suggests that driving skills and experience may not necessarily result in improved neural conduction efficiency in the frontal cortes, which are responsible for higher cognitive processes. The frontal cortex generally maintains relatively high structural stability after the age of 30, especially when compared to sensory-related brain regions (*Bethlehem et al., 2022*; *Blakemore, 2012*). Since our participants were mostly between 30 and 60 years old, it was expected that the frontal cortex would not exhibit changes in gray matter volume with driving skills and experience.

Previous task and resting state fMRI studies have indicated the involvement of the frontal cortex in the driving-related skill (*Calhoun et al., 2002*; *Spiers & Maguire, 2007*; *Wang et al., 2015*). These results suggest that the frontal cortex continue to actively participate in driving-related cognitive activities. Combining our findings of no significant driving-related changes in frontal gray matter volume in this study, we posit that driving skills and experience may lead to a more active transfer of information from the sensory cortex to the frontal cortex. However, this does not necessarily result in more efficient road scenario analysis, more decisive and accurate decision-making, or more precise execution control activities. This finding may help explain why experienced drivers still exhibit a considerable accident rate.

It is noteworthy that, despite employing experienced drivers in a complex urban setting as participants, we did not observe hippocampal changes similar to those reported by *Maguire et al. (2000)*, *Maguire, Woollett & Spiers (2006)*, specifically the increase in posterior hippocampal volume and decrease in anterior hippocampal volume. *Megías et al. (2018)* also attempted to replicate the results of Maguire et al. and, like us, they failed to find significant differences in hippocampal gray matter volume between drivers and non-drivers. Therefore, we argue that the results from Maguire et al. should undergo further validation.

The current preliminary findings highlight a relationship between driving experience and brain area volumes, yet longitudinal studies are needed to establish causal evidence for the dynamic changes in brain structures associated with driving. Moreover, larger sample sizes are essential to draw more robust conclusions. Additionally, investigating how the

brain volume of drivers varies across different geographical environments, such as mountainous *vs* flat city roads in places, presents an intriguing avenue for future research (*Coutrot et al., 2020*; *Kühn et al., 2017*, *2020*). Finally, considering the role of gene-environment interactions in shaping brain structure, incorporating twin studies could further enhance our understanding of this issue.

## CONCLUSION

This study employed VBM to investigate structural differences in the brains of drivers and non-drivers, offering insights into how driving experience contributes to more efficient driving-related cognitive processing. Gray matter volume measurements revealed that, in comparison to non-drivers, drivers exhibited significantly smaller gray matter volumes in the middle occipital gyrus, middle temporal gyrus, supramarginal gyrus and cerebellum_4_5, indicating an association with driving skills. However, no such changes were detected in the frontal lobe associated with driving skills or experience. Further correlational findings suggested a potential reduction in gray matter volume in occipitotemporal regions with increasing driving experience. This suggests that the enhanced efficiency in driving-related activities and improved sensory processing efficiency associated with driving experience may not be attributed to increased efficiency in higher-level cognitive processing activities related to the frontal cortex. The current findings support the myelination theory.

## ACKNOWLEDGEMENTS

We would like to thank all participants who took part in the present study.

### Funding

This research was supported by grants from the National Natural Science Foundation of China (NSFC31970989). The funders had no role in study design, data collection and analysis, decision to publish, or preparation of the manuscript.

### Grant Disclosures

The following grant information was disclosed by the authors:
National Natural Science Foundation of China: NSFC31970989.

### Competing Interests

The authors declare that they have no competing interests.

### Author Contributions

- Jiangtao Chen conceived and designed the experiments, performed the experiments, analyzed the data, prepared figures and/or tables, authored or reviewed drafts of the article, and approved the final draft.
- Xiaoyu Chen conceived and designed the experiments, authored or reviewed drafts of the article, and approved the final draft.

- Li Gong performed the experiments, authored or reviewed drafts of the article, and approved the final draft.
- Di Zhang performed the experiments, authored or reviewed drafts of the article, and approved the final draft.
- Qiang Liu conceived and designed the experiments, authored or reviewed drafts of the article, and approved the final draft.

### Human Ethics

The following information was supplied relating to ethical approvals (*i.e.*, approving body and any reference numbers):

Southwest University Brain Imaging Center Institutional Review Board

### Data Availability

The raw data are available in the Supplemental File.

### Supplemental Information

Supplemental information for this article can be found online at http://dx.doi.org/10.7717/peerj.17228#supplemental-information.

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
