# Peer review of "Behind the wheel: exploring gray matter variations in experienced drivers"

_PeerJ, doi:10.7717/peerj.17228_

## Round 0.1 · original submission · Major Revisions

Dear Authors,

A major revision is needed for this manuscript. Please revise as per comments from the 3 reviewers.

Reviewer 1 ·

Basic reporting

- In table 2, p-values should be added to elucidate the presence or absence of significant differences between the two groups across various outcomes.

- In table 3, the term "BA" is used in the text without a clear explanation. It would be helpful for the authors to elaborate on the meaning of "BA" and its relevance within the context of the study.

- In Figure 1, two subplots are duplicated

Experimental design

- In line 120-121, the authors noted that participants were matched according to age and educational backgrounds. The authors can provide more details on how the matching process works.

Validity of the findings

- Comparing the data presented in Table 3 with that in Figure 2, inconsistencies are noted in the reported p-values. Specifically, for the variable MOG_R, the p-value is indicated as 0.029 in Table 3, whereas in Figure 2, it is denoted as smaller than 0.01. This disparity raises questions about the accuracy and coherence of the statistical reporting, and it would be beneficial for the authors to provide clarification on this discrepancy.

- In section 3.3, when exploring the relationship between the volume of ROIs and the duration of driving experience, is age and other demographic factors adjusted in calculating the correlation similar to the group-wise comparison in GM volumes? The incorporation of adjustments for potential confounders, including demographic variables, holds the potential to strengthen the credibility for the causal relationship between ROI volume and the duration of driving experience.

- In the discussion section, the authors can add some discussion points on mentioning the limitation of the studies. Brain structure is also impacted by other factors such as genetics and environmental factors. Given the small sample size of the study, it should be cautious to generalize the observed association to a more extensive population.

Reviewer 2 ·

Basic reporting

no comment

Experimental design

no comment

Validity of the findings

1. The stats for Global volume analysis are missing. (Line175)


2. The authors claimed significantly lower volume for the driver, but the statistical results need to be corrected for multiple comparisons. (Line 183) The voxel-based analyses contain a huge number of comparisons which might lead to a large number of false positives, therefore multiple comparison correction is necessary.


3. The authors observed significant correlations between brain area volume and driving years, which implies the association between driving-induced brain structural changes. However, it might be confounded by the age of the driver since age and driving years are correlated too. As a result, age should be included in the model to rule out such possibility.

·

Basic reporting

1) Update the caption of Table 3 to reflect that it presents results for both gray matter (GM) and white matter (WM), among other variables.

2) Clarify the definition of ROIs (Regions of Interest) in line 181 to avoid any contradiction with the findings in lines 175 and 176.

3) For Figure 2, consider displaying standard deviation (std) instead of Standard Error of the Mean (SEM) to provide a more accurate representation of data distribution.

4) Specify whether the observed changes in the results are positive or negative for better clarity and interpretation.

5) Is there any WM related findings in drivers to support myelination theory?

Experimental design

In my review of the paper, a notable concern arises in Table 1, specifically concerning participants' demographics. The range in the Drivers group (standard deviation = 6.07) appears significantly larger than in the Non-drivers group (standard deviation = 1.89). Despite the absence of significant differences in age between the two groups, it is crucial to consider age as a potential confounding factor. This becomes particularly pertinent when acknowledging the negative association between age and gray matter volume in middle-aged participants. Hence, I recommend that the analysis incorporates age as a confounding factor to ensure the robustness and accuracy of the study's findings.

Validity of the findings

Given the negative association between age and gray matter volume in middle-aged participants, the failure to address age as a potential confounding factor is a notable oversight. This omission has implications for the validity and reliability of the study's conclusions. To align with our standards, it is imperative that the analysis explicitly incorporates age as a confounding factor. This adjustment will enhance the robustness and accuracy of the findings, ensuring a more rigorous and valid contribution to the field.

---

## Round 0.2 · Minor Revisions

Please sort out the claims mentioned by one of the peer reviewer based on your p value

Reviewer 1 ·

Basic reporting

I think that the authors have adequately addressed the comments made by the reviewers in the revised version of the manuscript. Therefore, I have no further comments.

Experimental design

I think that the authors have adequately addressed the comments made by the reviewers in the revised version of the manuscript. Therefore, I have no further comments.

Validity of the findings

I think that the authors have adequately addressed the comments made by the reviewers in the revised version of the manuscript. Therefore, I have no further comments.

Reviewer 2 ·

Basic reporting

no comment

Experimental design

no comment

Validity of the findings

The authors concluded significant negative correlations of grey matter volume in the right middle occipital gyrus and left middle temporal gyrus with driving years, while the statistical test found approaching significance of p=0.075 and p=0.078. The authors at least need to justify the strength of the evidence for making the claims.

Additional comments

no comment

·

Basic reporting

Authors have addressed all the required comments.

Experimental design

Authors have addressed all the required comments.

Validity of the findings

Authors have addressed all the required comments.

Additional comments

Authors have addressed all the required comments.

---

## Round 0.3 · accepted · Accept

Thank you for your submission that has been accepted and will go through editorial/ galley proof processes.

Reviewer 2 ·

Basic reporting

no comment

Experimental design

no comment

Validity of the findings

no comment